# SF-Partition-Based Clustering and Relaying Scheme for Resolving Near–Far Unfairness in IoT Multihop LoRa Networks

**DOI:** 10.3390/s22239332

**Published:** 2022-11-30

**Authors:** Dick Mugerwa, Youngju Nam, Hyunseok Choi, Yongje Shin, Euisin Lee

**Affiliations:** 1School of Information and Communication Engineering, Chungbuk National University, Cheongju 28644, Republic of Korea; 2Research Institute for Computer and Information Communication, Chungbuk National University, Cheongju 28644, Republic of Korea

**Keywords:** long range (LoRa), spreading factor (SF), lower spreading factor zone (LSFZ), higher spreading factor zone (HSFZ)

## Abstract

Long range (LoRa) is one of the most successful low-power wide-area networking technologies because it is ideally suited for long-distance, low-bit rate, and low-power communications in the unlicensed sub-GHz spectrum utilized for Internet of things (IoT) networks. The effectiveness of LoRa depends on the link budget (i.e., spreading factor (SF), bandwidth (BW), and transmission power (TX)). Due to the near–far effect, the allocation of a link budget to LoRa devices (LDs) in large coverage regions is unfair between them depending on their distance to the GW. Thus, more transmission opportunities are given to some LDs to the detriment of other LD’s opportunities. Numerous studies have been conducted to address the prevalent near–far fairness problem. Due to the absence of a tractable analytical model for fairness in the LoRa network, however, it is still difficult to solve this problem completely. Thus, we propose an SF-partition-based clustering and relaying (SFPCR) scheme to achieve enormous LD connectivity with fairness in IoT multihop LoRa networks. For the SF partition, the SFPCR scheme determines the suitable partitioning threshold point for bridging packet delivery success probability gaps between SF regions, namely, the lower SF zone (LSFZ) and the higher SF zone (HSFZ). To avoid long-distance transmissions to the GW, the HSFZ constructs a density-based subspace clustering that generates clusters of arbitrary shape for adjacent LDs and selects cluster headers by using a binary score representation. To support reliable data transmissions to the GW by multihop communications, the LSFZ offers a relay LD selection that ideally chooses the best relay LD to extend uplink transmissions from LDs in the HSFZ. Through simulations, we show that the proposed SFPCR scheme exhibits the highest success probability of 65.7%, followed by the FSRC scheme at 44.6%, the mesh scheme at 34.2%, and lastly the cluster-based scheme at 29.4%, and it conserves the energy of LDs compared with the existing schemes.

## 1. Introduction

In the dawn of the new era of the Internet of things (IoT), the fast development of wireless communications and device technologies enables the interconnection and exchange of data among sensors. In the near future, the IoT will have an extraordinary impact on humans as a result of an increase in the ubiquity of the Internet by integrating every object with a motive of interaction via embedded systems in many fields such as asset tracking, agriculture, smart homes, and smart cities [1].

There are numerous predictions being made about the growth of connected devices.

According to recent studies [2], every person will be expected to have at least six IoT devices, with a monthly global mobile traffic of 77 exabytes. Usually, the devices are expected to cover a large geographical area for a longer span of five to ten years and send interesting data of small size, such as humidity, temperature, and other variables around them, over a longer distance using wireless communications.

For reflecting the above requirements, low-power wide area networking (LPWAN) technologies with a low cost, low bandwidth, and a low per-unit consumption have been developed for ubiquitous IoT connectivity of a large geographical area [3], different from conventional IoT networking technologies such as ZigBee and Bluetooth based on a shorter communication range. As representative protocols of LPWAN, LoRa, Sigfox, RPMA (Random Phase Multiple Access), and the Weightless protocol have been developed for LPWAN-based IoT implementation within the unlicensed ISM bands. Among them, LoRa is one of the most successful LPWAN technologies because it is ideally suited for long-distance, low-bit rate, and low power communications. LoRa communication technology was first proposed by SemTech and is now being developed by LoRa^TM^ Alliance [4]. LoRa is a physical layer technology that uses a proprietary spread spectrum technique to modulate signals in unlicensed sub-GHz ISM bands (868 MHZ in Europe, 915 MHZ in North America, and 433 MHZ in Asia).

LoRa provides bidirectional communication via the chirp spread spectrum (CSS) modulation, which spreads a narrow-band signal over a wider channel bandwidth, making it more interference resistant [5]. LoRa primarily employs six spreading factors (SF) ∈{7,8…,12} to adapt to the trade-off between the data rate and the communication range. In fact, the greater the SF, the greater the range with a lower data rate, and vice versa [6]. The data rates vary with the SF and channel bandwidth, ranging from 300 bps to 50 kbps. According to [7], LoRa technology is based on a star of stars topology, in which GWs relay messages between end devices and a central network server. This model is distinguished by the provision of a long-distance and reliable link via a special modulation technique, in which a LoRa GW collects raw data directly from end devices by single-hop communications and forwards it to a network server (NS) connected by a high-speed backhaul network as shown in Figure 1 [8].

LoRaWAN is a LoRa MAC layer that employs the ALOHA protocol [9], which is managed by an NS [10]. Furthermore, LoRaWAN defines three types of devices with capabilities (classes A, B, and C). Class A devices use pure ALOHA access for the uplink, are battery limited, and can only receive an ACK in two receiving windows during downlink transmission. Class B devices, in contrast to class A, are beacon-based and can open an additional receiving window at predetermined times. Except when transmitting, class C devices are always listening to a channel and consume a lot of energy [11]. The three classes can coexist on the same network, and devices can switch between them. Because of the orthogonality of the sub-bands and the quasi-orthogonality of different SFs, a LoRa GW can receive packets from multiple LoRa devices at the same time. The LoRa Alliance has defined the higher layers and network architecture known as LoRaWAN, with the medium access control protocol (MAC) layer essentially being an ALOHA variant of random access.

Generally, a massive number of LoRa devices may be connected to a GW because of the vast coverage area factor, which creates a near–far fairness problem due to the heavy route loss as the bottleneck in overall system performance in LoRa networks. Moreover, LoRa networks also suffer from collisions of concurrent transmissions on the same channel and SF in closely packed deployment situations. In this regard, the conventional protocol for pure ALOHA in IoT LoRa networks is proven to be oversimplified because it fails to account for channel fading, power control, and aggregate interference [12]. To overcome this issue in LoRa, previous studies of [13,14,15] proposed multihop communication schemes using relay devices and a programmed e-node that acted as a range extender to increase the reliability of data forwarding from LoRa devices with duty-cycle constraints to a GW in IoT LoRa networks. In this case, class C devices acted as relay devices, intercepting data transmissions from LoRa devices by overhearing and relaying them to the GW. Furthermore, each relay device alternated between overhearing the end devices’ transmissions during the receive window and forwarding the contents of the overheard packets to the GW on a regular basis. However, these studies focused on only increasing coverage extension probability while ignoring fairness in terms of network area success probability.

### 1.1. Background and Motivation

To solve the shortcomings of the existing studies, we propose an SF-partition-based clustering and relaying scheme to address the near–far unfairness problem caused by the low success probability, long channel occupancy time (also known as time on air (ToA)), redundancy, low coverage, capture effect challenge, and collisions in IoT LoRa networks. For this, the proposed SFPCR scheme has three methods: SF-based network partitioning, density-based subspace clustering, and relay LD selecting. The SF-based network partitioning method is used to determine the appropriate partitioning threshold point for bridging the packet delivery success probability gap between different SF regions in an IoT LoRa network. Then, the packet delivery success probability is evaluated based on the hop distance of each hop in multihop LoRa communications. By the partitioning threshold point, the IoT LoRa network is separated into two SF zones: the lower SF zone (LSFZ) and the higher SF zone (HSFZ). To avoid long-distance transmissions with low reliability from individual LDs in the HSFZ of the GW, the density-based subspace clustering method creates arbitrary-shaped clusters by grouping adjacent LDs in the HSFZ and selects the CH of each cluster for gathering and transmitting data packets of cluster members. For clustering the arbitrary-shaped clusters and choosing CHs based on the CH conditional probability, a combination of the DBSCAN algorithm and the naive Bayes classifier is used. To support reliable data transmissions from each CH in the HSFZ to the GW by multihop communications, the relay LD selecting method chooses the best relay LD in the LSFZ to extend uplink transmissions from the CH to the GW. Then, the best relay LD is determined by the harmonic mean of the packet success probability and the remaining energy selection strategy of all candidate LDs in the LSFZ.

### 1.2. Contribution

In order to achieve massive scalability and connectivity, we aim to maximize the minimum success probability of all distant LDs and optimize SF allocation in all SF zones of the IoT LoRa network within a predetermined duty-cycle limitation.

The detailed contributions of this paper can be summarized as follows.

We provide an SF-based network partitioning method to determine the best partitioning threshold point for dividing SF zones of an IoT LoRa network into the LSFZ and the HSFZ. To bridge the performance gap among the SF zones, an optimal SF division method based on a heuristic algorithm is proposed.We provide a density-based subspace clustering method for grouping adjacent LDs in the HSFZ. To construct clusters of arbitrary shape and to select CHs by a binary score representation, a combination of the DBSCAN algorithm and naive Bayes classifier is proposed.We provide the relay LD selecting method to ideally choose the best relay LD only in the LSFZ for extending transmissions of CHs in the HSFZ to the GW by multihop communications. The harmonic mean based on the packet success probability and the remaining energy is proposed to choose the best one among all candidate relay LDs.We maximize the performance of packet success probability, scalability, and fairness in the SF zones of an IoT LoRa network. To ensure the optimal zone allocation with connection fairness, the packet success probability of each LD in a certain SF zone is examined.We conduct simulations in various environments for evaluating the performance of the proposed SFPCR scheme. Simulation results demonstrate that the proposed SFPCR scheme raises the packet success probability by bridging the packet delivery success probability gap between SF zones while conserving more energy among LDs by reducing the number of packets compared with existing schemes.

### 1.3. Organisation

The remainder of this paper is organized as follows. First, we provide the related works on the proposed scheme in Section 2. The system model of the proposed scheme is presented with the network, energy, and success probability models in Section 3. Our SF-partition-based clustering and relaying scheme is described in Section 4. The performance evaluation and numerical results are presented and discussed in Section 5. Finally, the paper is concluded in Section 6.

## 2. Related Works

Recently, numerous overviews on LoRa technology and LoRaWAN networks have been published [16,17]. They deal with various research fields such as network lifetime, spreading factor allocation, energy conservation, coverage, connectivity, and performance improvement. LoRa technology has grown in popularity in a variety of industries, including smart metering [18], smart cities [19], fleet tracking [20], and energy management. IoT applications necessitate devices transmitting over long distances, which exposes signals to interference from various sources. Similarly, LoRaWAN coverage and scalability have been studied both indoors and outdoors to achieve a coverage range of 5 to 15 km in an open area with limited interference [16]. However, due to the changing environment, these studies on interference have not fully mitigated the problem to improve the transmission success probability. Unlike the Sigfox communication model, LoRa uses chip spread spectrum (CSS) modulation, which allows for a higher receiver sensitivity and thus a greater resistance to interference. However, there is still a lot of research being conducted on how to reduce cospreading factor interference and collisions caused by two or more LoRa devices transmitting their packets at the same SF [6]. Many other works assume that the SFs are static and are assigned based on the distance between the LoRa devices and their nearest GW, forming a kind of tiered ring structure around each GW. To solve the CSS modulation problems of LoRa communications, there are three research areas: spread factor allocation, clustering, and multihop communication. Thus, we examine the three research areas in the next three subsections in detail, respectively.

### 2.1. Spread Factor Allocation in LoRa Networks

The performance of LoRa networks has been aided by appropriate resource allocation strategies, one of which is SF, which defines the number of bits that can be encoded by a symbol. SF assignment schemes include, but are not limited to, random, area-proportional, and distance-based distribution schemes from the GW [10,21]. Furthermore, SF assignments in the LoRa network are more attributable to the work in [22], which indicated that SF allocation was primarily based on the power level of the signal that the GW receives from the end devices and the GW’s sensitivity without taking end device location into account. Similarly, in [23], an efficient interference-aware SF allocation strategy was introduced. In reality, SF can be assigned based on the signal-to-noise ratio (SNR) of the received packets, and as a result, devices in a certain range can use the same SF [24]. However, the challenge to optimize the SF assignment over each channel realization and high-density building obstacles was ignored. In [25], the authors considered a fair SF clustering of end devices in LoRa networks using k-means clustering. Their algorithm determined the minimum cost function, which was defined as the distance between each point in the data set and its nearest centroid. Furthermore, the k-means clustering method achieved robust clustering with ease. However, this method of SF allocation did not take into account the SF overlap of edge nodes at different annulus boundaries as well as the distance of the end devices from the GW. Studies in [26] proposed an SF allocation algorithm based on matching theory to maximize the minimum achievable average rate in LoRa networks to improve network throughput and fairness. This theory classifies resource allocation as a many-to-one matching problem with conventional externalities and peer effects. In this case, a device prefers to be paired with the SF that provides the most utility. That algorithm, however, performed better with only heterogeneous LoRa devices. Reynders et al. [27] proposed a heuristic SF allocation scheme in which users with similar path losses were simply assigned to the same channel, then to each SF based on their distance from the GW. The work in [28] proposed a suboptimal SF allocation strategy to maximize packet success probability. Inter-SF interferences, on the other hand, were ignored in that work. In particular, the proposed SFPCR scheme allocates SF based on a combination of the device’s distance and the SNR of the received packets. The SF allocation technique utilized in the SFPCR scheme is modified by assuming that all LoRa devices inside the same contour lines have the same SF.

### 2.2. Clustering in LoRa Networks

Clustering is one of the useful methods for enhancing network connectivity and prolonging the network lifetime in wireless networks by managing nodes efficiently. In clustering for IoT LoRa networks, LoRa devices are grouped into clusters, and the cluster head (CH) in each cluster acts as the mediator for data transmissions between LoRa devices and a GW. Clustering not only reduces packet redundancy but also manages efficient data routing, resulting in improved efficiency, scalability, and the avoidance of redundant message exchange. Clustering protocols enable end devices to reduce data packets on networks through data aggregation, which improves network lifetime and energy consumption. As a result, several studies based on clustering algorithms have been proposed for IoT LoRa networks. In [29], the authors proposed a two-stage energy-efficient cluster-based solution for data collection by a mobile sink in a robot network by considering the first stage as a travelling salesman problem and the second stage as the removal of some CH robots from the path to reduce the UAV’s energy consumption. In [30], the cluster-based layering approach for uplink multihop communication was analysed. In that paper, the authors presented a multihop uplink communication scheme that took advantage of LoRa’s ability to customize its transmission parameters and combines them with a novel routing protocol to solve network coverage and high energy consumption problems. Clusters were used in that scheme to associate an end device with exactly one GW, and once an end device was associated with a GW, the end device communicated through the GW. This type of scheme, however, limited the ability of end devices closer to the GW to transmit directly to the main GW. In [21], an SF-based clustering scheme was proposed to improve the multihop capacity of the LoRa network. Based on network clustering, each subnet rooted at a sink node was assigned a specific SF. The procedure focused on balancing airtime between subnets while maintaining connectivity.

However, end device extraction from a higher SF subnet has an impact on network maintenance and appears to be an unrealistic scenario. Based on this clustering, Ref. [31] described a method for logically partitioning end devices into several clusters. End devices in nearby clusters served as relay devices for end devices in the next more distant zone. As a result, clustering with relaying provided an appealing solution to this problem by focusing on distant end devices one at a time. Clusters formed in that case were either the same or different widths. The clustering style in that approach considered clusters of the entire network but ignored the transmission capability of end devices closer to the GW.

As examined above, the existing studies concentrate on clustering techniques that partition the entire network in order to reduce energy consumption and improve connectivity between LoRa devices. However, these approaches deprive LoRa devices with a high link quality of directly connecting to the GW and acting as relay LoRa devices of distant devices. As a result of only forming subspace clusters in the designated HSFZ, among the benefits of the proposed density-based subspace clustering are a reduced energy consumption, communication overhead, and increased connectivity.

### 2.3. Multihop Communication in LoRa Networks

Multihop communications are used to extend the connection distance over the communication range of single-hop communications. To enable coverage extension in IoT LoRa Networks based on single-hop communications, many studies have been published in IoT LoRa networks. The first multihop LoRa network was investigated in [32]. As a result, when a data packet was broadcast, all neighbouring end devices checked its origin in terms of hop distance and relayed the messages if they were closer to the sink. The algorithm allowed the use of beacons for time synchronization as well as for the communication distance between each end device and the GW. However, it caused a lot of packet redundancy and delay, which slowed down network performance. Aslam et al. [19] investigated the use of multihop LoRa topologies to enable energy-efficient connectivity in smart city applications. In this study, packet reception rates for various source-to-destination distances, SFs, and transmission powers were used to evaluate single-hop and multihop LoRa topologies. Then, the results of the study demonstrated that two-hop networks could significantly outperform single-hop networks in terms of range. However, that study did not go into detail about the best relay placement to influence range extension for distant end devices and the powerful impact. In [33], the authors proposed a multihop communication scheme called FSRC based on a selective relay operation. By controlling communication parameters, the relay control strategy of the FSRC maximized both coverage probability and minimum success probability for all SF regions. Like other similar schemes [34,35], they introduced a programmed e-node and an implicit relay node to act as a transparent range extender with an overhearing operation, respectively. With the primary goal of replicating the same packet sent by distant end devices, they received it and relayed it to the GW by overhearing. However, these implementations generated a large amount of network traffic from broadcasts, resulting in performance degradation.

In [36,37], an energy-efficient multihop communication scheme called e2MCH and a LoRa mesh networking system for large-area monitoring of IoT applications were studied. The research presented in these papers introduced multiple relay devices for collecting data from IoT sensors spread across a large geographical area and forwarding them to a GW in close proximity. The research, however, was vulnerable to packet collisions, which reduced the reliability and energy efficiency of LoRa networks. The proposed SFPCR scheme employs a centralized optimum relay assignment scheme that iteratively chooses the best relay LoRa device from the candidate set of devices in the LSFZ based on both packet success probability and residual energy, to transmit a data packet from the subspace clusters in the HSFZ. Consequently, the proposed multihop approach offers a practical framework for reducing the transmission distance and transmission power required by distant nodes with large spreading factor values.

## 3. System Model

In this section, we provide a detailed description of the system model in the proposed scheme. We first present the network model for deploying an IoT LoRa network. Next, we derive the energy model and the success probability model for LoRa devices in various spreading factor zones based on the network model, respectively.

### 3.1. Network Model

As the IoT LoRa network shown in Figure 2, we considered a classic uplink LoRa system with a single GW at the centre of a network field of radius R=6 km and a set of a lot of LoRa devices (LDs), which was defined by $LD=n_{i}|I\in \;\left\{1,\ldots ,N\right\},$N=|LD|. LoRa devices were allocated in Gaussian distribution within the network field. For the proposed SFPCR scheme, we divided the IoT LoRa network into two regions: a low spreading factor zone (LSFZ) and a high spreading factor zone (HSFZ). In our network model, a LoRa device located in the HSFZ joins the CH in a cluster through a single hop communication and sends its data packet to the CH. Then, the CH forwards data packets from LoRa devices in its cluster to a selected relay LoRa device in the LSFZ, which transmits these data packets in a single-hop communication to the GW, as shown in Figure 2.

The performance of LoRaWAN is restricted by a duty cycle in a sub-band for the amount of time spent transmitting packets, which is referred to as the time on air (ToA). As a consequence of this, we took into consideration a duty cycle ($\delta_{i}$) of 1% [4], operating at a certain frequency (Fc) in the 868 MHz band with a code rate (CR) of 4/5 and a variable bandwidth (BW) ranging from 125 kHz to 500 kHz. We explicitly deployed class A and B LoRa devices, with LoRa devices always initiating transmission. During a time slot, each LoRa device $n_{i}$ with matching SF ∈  {7, 8, …, 12} and values for signal-to-noise ratio (SNR), receiver sensitivity for a specific bandwidth, and bit rate, as shown in Table 1, transmits a packet. The LoRa devices use different SFs for transmission to ensure orthogonality and multiple detections at the receiver.

In [38], the authors alluded that different SFs caused significantly different times on air for symbol transmission. A symbol can encode *k* information bits into a chirp with SF=k, and the bit rate $R_{b}$, so the symbol period is evaluated by $T_{symbol}=\frac{2^{k}}{BW}$, where BW is the bandwidth. When SF=k+1, one symbol period is $\frac{2^{n+1}}{BW}$, which doubles the transmission time by sending only one more bit. However, a higher SF indicates a greater resistance to interference and noise, resulting in a greater communication range.

According to recent research, we assumed that an IoT LoRa network was divided into multiple annular rings of the same width. Thus, the width of the annulus was defined here as the Euclidean distance between the two closest points on the inner boundary $C_{in}$ and the outer boundary $C_{out}$, respectively. Recent work has shown a spreading factor distribution scheme based on a certain range of distances to the GW. According to [22,39], the spreading allocation is primarily based on the power level of the signal that the GW receives from the devices and the GW sensitivity without taking the device location into account. Consequently, it is not trivial to consider a general coverage contour metric to determine the degree of coverage at any point within the IoT LoRa network while avoiding interference from obstacles. In contrast, the proposed SFPCR scheme determines and allocates the appropriate SF based on the transmission distance and SNR of the received packets, as described in [24]. In addition, the SF allocation technique utilized in the SFPCR scheme was modified by assuming that all LoRa devices inside the same contour lines had the same SF. Due to obstructions and interference, LoRa devices may be the same distance apart but have different SNR values. Therefore, these devices cannot share an SF.

### 3.2. Energy Model

For the energy model of LoRa devices in the proposed scheme, we employed the same energy model as [38,40]. Unlike class C LoRa devices, which are usually mains-powered, the energy consumption of each LoRa device must be precisely stated. The energy consumption $E_{tx,i}$ for transmitting a β-bit LoRa packet from a LoRa device LD(i) to the GW can be categorized as follows: the LoRa device waking up, low-power listening, radio preparation, signal transmission, radio off, and processing. Furthermore, the consumed energy for signal transmission is affected by the transmission power as well as the varying transmission time caused by the spreading factor. It is precisely defined as (Equation 1) [41],
(1)$$E_{tx,i}=P_{tx}\times T_{SF(i)}$$
where $P_{tx}$ denotes the transmit power with time on air for a LoRa device with *SF*(*i*), and $T_{SF(i)}$ denotes the time on air for LD(i) expressed as (Equation 2) and (Equation 5) [38].
(2)$$T_{SF(i)}=T_{symbol}\times \left(N_{pr}+N_{pl}\right)$$
(3)$$N_{pr}=\left(n_{preamble+4.25}\right)\times \frac{2^{SF_{i}}}{BW}$$
(4)$$N_{pl}=8+\max\left(\left[\frac{8L-4SF_{i}+44}{4(SF_{i}-2DE)}\right]CR,0\right))\times \frac{2^{SF_{i}}}{BW}$$
(5)$$T_{SF(i)}=\left(20.25+\max\left(\left[\frac{8L-4SF_{i}+44}{4(SF_{i}-2DE)}\right]CR,0\right)\right)\times \frac{2^{SF_{i}}}{BW}$$

In Equation (Equation 2), $N_{pr}$ and $N_{pl}$ represent the number of symbols in the packet preamble and payload, respectively. Substituting for $T_{symbol}$, $N_{pr}$, and $N_{pl}$ in (Equation 2), we get (Equation 5), if low data rate optimization is enabled, DE=1, otherwise DE=0, CR =4/5 represents the code rate, and $n_{preamble}=8$ is the preamble length. It is worth noting that the energy consumption of a LoRa device in the annulus (*j*) is proportional to airtime (τ
*j*), with large values for a high SF.

Subsequently, we evaluated the energy consumption $E_{n-hops}\left(SF,P_{tx}\right)$ using the multihop communication from a LoRa source device(i) to the GW through n hops. Then, the energy consumption for sending a packet was computed as follows:(6)$$E_{n-hops}\left(SF,P_{tx}\right)=\sum_{i=1}^{n}\left(E_{tx,i}+E_{rx,i}\right)$$
where $P_{tx}$ is the consumed power according to the supply current and supply voltage for the transmitter, $E_{tx,i}$ and $E_{rx,i}$ are the energy consumption of the transmitter and receiver in the *i*th hop according to its current configuration on the transmitted power, spreading factor, and bandwidth, respectively.

Classs A and B LoRa devices on the LoRa network typically spend the bulk of their time in light-sleep mode due to their low energy consumption and battery power. Meanwhile, in order to calculate the total dissipated energy consumption $E_{total\_N_{i}}\left(t\right)$ of a LoRa device $N_{i}$ at a specific time (t), we considered two states, active and inactive modes, given as (Equation 7) and (Equation 8) and thus derived $E_{total\_N_{i}}\left(t\right)$ as follows.
(7)$$E_{total\_N_{i}}\left(t\right)=E_{inactive\_N_{i}}\left(t\right)+E_{active\_N_{i}}\left(t\right)$$

Here, $E_{inactive\_N_{i}}\left(t\right)$ is the dissipated energies in the state where devices are in sleep mode ($E_{sleep\_N_{i}}\left(t\right)$) relative to 1.8 μA [42] and $E_{active\_N_{i}}$ is the dissipated energies by the LoRa device during the transmission mode, reception mode, data measurement, processing, and wake up. Thus, $E_{active\_N_{i}}$ can be calculated as follows:(8)$$\begin{array}{c} E_{active\_N_{i}}\left(t\right)=E_{Wu}+E_{proc}+E_{Wut}+\sum_{j=1}^{n}E_{s}\left(i\right) \end{array}$$
where $E_{Wu}$ is the energy consumption when a device wakes up, $E_{proc}$ is the energy consumption of data processing, and $E_{Wut}$ is the amount of energy to move the transceiver from sleep mode to active mode, respectively. In particular, $E_{s}\left(i\right)$ is the energy consumption of a LoRa device in terms of data rate to transmit $L_{packet}$ with *n*-hop communication evaluated as (Equation 9) [43] and is calculated as follows:(9)$$E_{s}\left(i\right)=\frac{L_{packet}}{R_{b}}\times \left(P_{tx}+P_{rx}\right)$$
where $L_{packet}$ is the size of the transmitted packet in bits, $P_{tx}$ and $P_{rx}$ are the consumed power according to the supply current and supply voltage for transmitter and receiver, respectively, and $R_{b}$ is the bit rate defined in Equation (Equation 10).
(10)$$R_{b}=\frac{CR\times SF}{\frac{2^{SF}}{BW}}$$

Here, SF is the spreading factor, BW is the bandwidth, and CR is the code rate defined as $\frac{4}{\left(\iota +4\right)}$ with ι∈{1,2,3,4}, respectively.

Ultimately, to understand the LoRa device’s energy consumption, it is essential to efficiently utilize the LoRa devices’ residual energy $E_{re\_N_{i}}\left(t\right)$ at a given time (t) for the effective selection of CHs and also prevent the untimely demise of devices to ensure that the energy of each LoRa device in the network is consumed in a relatively balanced manner. Thus, $E_{re\_N_{i}}\left(t\right)$ can be computed as (Equation 11),
(11)$$E_{re\_N_{i}}\left(t\right)=E_{0}-E_{total\_N_{i}}\left(t\right)$$
where $E_{0}$ is the initial energy of LoRa devices.

### 3.3. Success Probability Model

In this section, we examine the success probability of a signal in LoRa’s respective spreading factor regions under intra- and inter-SF interference. In this paper, we investigated the average success probability of LoRa devices in random locations in the field of an IoT LoRa network in order to determine the optimal number and positions of relay LoRa devices (LDs) in relation to the distance between a source LD and the GW. The proposed SFPCR scheme considers the evaluation of a source LD’s packet success probability using a varying SF to be dependent on the distance from the receiving LoRa device (i.e., the source LD (S) to the CH, the CH to the relay LD (R), and R to the GW). The collision of packets in a LoRa network is classified into two types: intra-SF collision and inter-SF collision [44]. If a transmission collides, the LoRa device becomes backlogged and slows down, resulting in excessive delay and packet loss. As a result, it is critical for the LoRa link to have a high data reliability.

Transmission from LDs located within a distance ≤r to the GW is considered successful in a LoRa network with a uniform distribution of N nodes over the disk of density $\lambda =\frac{N}{\pi r^{2}}$ if and only if a source LD (S) transmits a packet at distance x from the GW that is not affected by either intra-SF or inter-SF collisions. Furthermore, we assumed that LoRa devices released packets within predetermined time slots within a deployment radius of r, and that packet transmission intensity was $\theta =\frac{pkt}{\tau }$ (bits per second), where pkt is the number of packets and τ is the time interval in seconds.

Intra-SF interference occurs when LDs in the same spreading factor region and BW collide when the distance from the GW is less than xR. Here, *x* is a location of a LoRA node and $R=e^{\frac{6}{10\gamma }}>1$[45], and the density of the LDs within the same spreading factor region SF(i) is represented by $\rho_{i}=\frac{\alpha_{i}N}{\pi r^{2}}$ [44]. Subsequently, the number of potential interferers $I_{SF(i)}$ within the same spreading factor region (intra-SF) is represented by Equation (Equation 12).
(12)$$I_{SF(i)}=\alpha_{i}N\frac{{\left(min\left(xR,r\right)\right)}^{2}}{r^{2}}$$

Here, *N* is the number of LoRa devices in the disk, $\alpha_{i}$ is the percentage of LDs in a particular SF with $\sum_{7}^{12}\alpha_{i}=1$. Assuming that none of the potential interfering LDs initiate a transmission during the vulnerable period of $2T_{i}$ duration, $P_{suc}^{i}\left(loc\left(x\right)\right)$ is the probability of successful transmission of a LoRa device at location *x* in an IoT network area and is represented by the Equation (Equation 13),
(13)$$P_{suc}^{i}\left(loc\left(x\right)\right):=e^{-2T_{SF(i)}\theta I_{SF(i)}}$$
where $T_{SF(i)}$ is the time on air evaluated in Equation (Equation 2).

Consequently, the combined success probability $P_{suc}\left(S,GW\right)$ from the source LD (S) to the GW in the proposed SFPCR scheme takes advantage of the varying SF as defined in (Equation 14):(14)$$P_{suc}\left(S,GW\right)=P_{suc}^{SF(i)}\left(S,CH\right)\times P_{suc}^{SF(j)}\left(CH,R\right)\times P_{suc}^{SF(k)}\left(R,GW\right)$$

Here, S represents the source LoRa device $n_{i}$, CH is the cluster head represented as $CH_{i}$, R is the relay LoRa device also represented as $N_{i}^{*}$, and GW is the gateway, respectively. Then, $P_{suc}^{SF(i)}\left(S,CH\right)$ is the success probability from *S* to *CH*, $P_{suc}^{SF(j)}\left(CH,R\right)$ from CH to R, and $P_{suc}^{SF(k)}\left(R,GW\right)$ from R to GW. SF(i), SF(j), and SF(k) are the spreading factors assigned to each of them.

## 4. SF-Partition-Based Clustering and Relaying (SFPCR) Scheme

In this section, we explain the proposed SF-partition-based clustering and relaying (SFPCR) scheme in detail, which has three methods. First, we provide a detailed explanation of SF partitioning, followed by the density-based subspace clustering in the HSFZ and relay LoRa device selection mechanism, respectively.

### 4.1. SF-Based IoT Network Partitioning

In the SFPCR scheme, we divided an IoT LoRa network field into two zones, the low spreading factor zone (LSFZ) and the higher spreading factor zone (HSFZ), to resolve the near–far fairness problem in the packet success ratio. In the HSFZ, the LoRa nodes were clustered to enhance data collection ability and transmission reliability, and the cluster head (CH) in each cluster used a LoRa node in the LSFZ as a relay node to forward data to the GW. Algorithm 1 shows an SF partitioning algorithm to find an optimal partitioning threshold point ($P_{opt}$) for dividing an IoT LoRa network field into an LSFZ and an HSFZ. Then, a virtual partition threshold point $\left(\Theta_{th}\right)$ is used for making the LSFZ and the HSFZ. The algorithm employs a heuristic process with ($0\leq \Theta_{th}\leq 100$) until no packet success probability gaps exist. At each iteration, the algorithm evaluates the modified packet success probability of all the LoRa devices in the respective zones until $P_{opt}$ = $\Theta_{th}$. Moreover, if $P_{suc}^{i}\left(loc\left(x\right)\right)<\Theta_{th}$, the packet success probability of LDs in the HSFZ is being improved and the reverse is true for the packet success probability in the LSFZ.
**Algorithm 1:** Pseudocode for boundary cut-off point of LSFZ and HSFZ.**Input:**$P_{suc}^{i}\left(loc\left(x\right)\right)$: LoRa device success probability**Input:***N*: number of IoT LoRa devices**Input:***x*: LoRa device’s location**Input:** BW: bandwidth**Input:** SF: spreading factor of a LoRa device**Output:** Boundary point LSFZ and HSFZ, $P_{th}$1:Initialization2:Sum←03:$P_{max}\leftarrow 0$4:$P_{opt}\leftarrow 0$5:**for**$\Theta_{th}=0$: 100 **do**6:    **for**
*i*=1: *N* **do**7:        Calculate $P_{suc}^{i}\left(loc\left(x\right)\right)$ in Equation (Equation 13)8:        Calculate $P_{\Delta suc}^{i}\left(P_{suc}^{i}\left(loc\left(x\right)\right),\Theta_{th}\right)$9:        $Sum+=P_{\Delta suc}^{i}\left(P_{suc}^{i}\left(loc\left(x\right)\right),\Theta_{th}\right)$10:    **end for**11:    Calculate $P_{avg}\left(\Theta_{th}\right)$ = $\frac{Sum}{N}$12:    **if** $P_{max}$ <$P_{avg}\left(\Theta_{th}\right)$ **then**13:        $P_{max}$ = $P_{avg}\left(\Theta_{th}\right)$14:        $P_{opt}$ = $\Theta_{th}$15:    **end if**16:**end for**17:return
$P_{opt}$

Subsequently, the GW was placed at the network’s midpoint, and its location was used as a reference point to define the network’s zones. It should be noted that the optimal coverage range of each zone determined the LoRa devices density deployed in each zone. The initialization and beaconing processes provided the GW with global knowledge of the entire network. Thence, we deduced the average success probability $P_{avg}\left(\Theta_{th}\right)$ for all LoRa devices, necessitating the determination of the partition threshold point $\Theta_{th}$ for the SF zones. Specifically, distinct virtual zones could be formed based on the network reliability deduced from the average success probability $P_{avg}\left(\Theta_{th}\right)$ for all the LoRa devices at a particular distance from the GW as shown in Equation (Equation 15) for respective SF among all nodes.
(15)$$P_{avg}\left(loc\left(x\right),\Theta_{th}\right)=\frac{1}{N}\sum_{i=1}^{N}\left(P_{suc}^{i}\left(loc\left(x\right)\right),\Theta_{th}\right)$$

In Equation (Equation 15), *N* is the number of LoRa devices whose packet success probability has already been registered at the GW. Under the condition that $P_{suc}^{i}\left(loc\left(x\right)\right)>P_{opt}$, LDs in the LSFZ have a success probability greater than the calculated threshold value. Here, $P_{opt}$ is based on the average success probability ($P_{avg}$) of all the LDs that take up the SF regions with faster chirps. Any LoRa device in the LSFZ, in particular, is a potential relay candidate for CHs. Otherwise, the remaining SF zone is referred to as the HSFZ. LoRa devices in the HSFZ have a lower success probability than the computed threshold if $P_{suc}^{i}\left(loc\left(x\right)\right)<P_{opt}$, which includes all LDs far from the GW and within the transmission radio range $\left(T_{r}\right)$ that suffer from near–far effect due to the fading channel and path loss caused by obstacles.

### 4.2. Density-Based Subspace Clustering in the HSFZ

Clustering is one of the main methods used to divide a network into a number of groups (clusters), with one node designated as the cluster head (CH) for each cluster [46]. Intuitively, a cluster in the SFPCR scheme is precisely defined as a set of density-connected LoRa devices in a particular region. Usually, the expected type of cluster is determined by the clustering criterion. The SFPCR scheme constructs LoRa nodes in the HSFZ as clusters. For clustering of the HSFZ, we used a combination of the DBSCAN algorithm [47] and naive Bayes classifier, a density-based subspace clustering algorithm that created arbitrary-shaped clusters and eliminated outliers, as shown in Figure 3. The naive Bayes classifier [48] was used for the CH selection based on the CH’s conditional probability. The DBSCAN clustering algorithm was suitable for the discovery of subspace clusters of arbitrary shape with a high efficiency in the network. The SFPCR scheme employs a centralized clustering strategy in which the network exerts complete control over the clustering procedure via the GW. In this method, the GW always selects which LD will function as CHs in the HSFZ, requiring periodic information on all LoRa devices to select the most suitable nodes. Moreover, there are eight standardized message types for LoRaWAN. We used the proprietary message type to encode a collection of messages necessary for network layering, cluster formation, and multihop message relaying. To facilitate proper cluster formation, we also defined a control message called the “HELLO” message. The GW periodically broadcasted the control message to all LDs within the cluster. The centralized clustering algorithm was dissected into two main phases: an initial setup phase and a steady-state phase for the subspace cluster formation and the CH’s selection criteria, respectively. We explain each of them in detail.

**Initial setup phase:** During this phase, the GW applies global knowledge from all the LoRa devices to build density-based subspace clusters based on the LoRa devices’ distribution and information. The SFPCR scheme generates density-based subspace clusters of LoRa devices whose neighbourhood contains a minimum number $\left(LD_{min}\right)$ of other devices within a given radius ($d_{0}$). In this case, $d_{0}$ is calculated using the *k*-nearest neighbour to its immediate neighbours [49]. Meanwhile, given the value *k* is represented as $LD_{min}$, we chose 4 as the *k* value, and the 4-nearest distance from LD(i) was the Euclidean distance from LD(i) to its 4-nearest neighbours, independent of the transmission range.Without loss of generality, three types of LoRa devices were defined: core LoRa devices, which contained at least $LD_{min}\geq 4$ LoRa devices [50] in their $d_{0}$ neighbourhood. Border LoRa devices did not have enough devices in their neighbourhood, but they were close to some core LoRa devices. Finally, other devices were considered outliers.Assuming a set of LoRa devices $N=\{N_{1},N_{2},\ldots N_{n}\}$ in the HSFZ made up of ψ subspaces ${\left\{C_{i}\right\}}_{i=1}^{\psi }\left(i=1,\ldots \psi \right)$, let $N_{i}$ be the set of $S_{i}$ LoRa devices belonging to subspace $C_{i}$ and $n=\sum_{i=1}^{\psi }S_{i}$. Starting with an arbitrary LoRa device $n_{i}$, the method returned all LoRa devices that were density-reachable from $n_{i}$ using $d_{0}$ and $LD_{min}$. As illustrated in Figure 3a, LoRa device $n_{j}$ is directly density-reachable from LoRa device $n_{i}$ if $n_{i}$ is a core LoRa device and $n_{j}$ is in its $d_{0}$ neighbourhood. A LoRa device $n_{j}$ is defined as density-reachable from a core LoRa device $n_{i}$ if there exists a chain of devices from $n_{i}$ to $n_{j}$, with each device being directly density-reachable from the prior LD. However, if $n_{i}$ is a border device, no devices are density-reachable from $n_{i}$ and then the algorithm visits the next LoRa device.To deduce the parameter $d_{0}$, we first computed the Euclidean distances with Equation (Equation 16) of all the $k(n_{i})$-nearest neighbours from Equation (Equation 17), and selected the maximum 4th -nearest neighbour distance $d_{0}$ as shown in the Equation (Equation 18).
(16)$$dist\left(n_{i},n_{j}\right)=\sqrt{{\left(X_{n_{i}}-X_{n_{j}}\right)}^{2}+{\left(Y_{n_{i}}-Y_{n_{j}}\right)}^{2}}$$
where $dist(n_{i},n_{j})$ is the Euclidean distance from the core LD $n_{i}$ to device $n_{j}$.
(17)$$k=min(LD_{max},LD_{min})$$In this case, $LD_{max}$ is the number of neighbours within an arbitrary subspace cluster distance ($d_{max}$), and $LD_{min}$ is the minimum number of LoRa devices to make a local subcluster, respectively.
(18)$$d_{0}=\underset{i}{\mathrm{argmax}}\;\left(dist\left(n_{i},n_{j}\right)\right)$$
for $n_{j}\in J_{i}$, where $J_{i}$ represents $n_{i}$’s nearest neighbours such that the number of LoRa devices in $\left(J_{i}\right)$ is equivalent to *k*. As a result, the algorithm merges all the local subclusters to form the arbitrary density-based subspace cluster based on ($k\left(n_{i}\right)=LD_{min}$) as shown in Figure 3b. However, if ($k\left(n_{i}\right)<LD_{min}$), no cluster is formed. A density connection can also refer to the relationship that exists between border LDs that are part of the same cluster but do not have a core LD in common with which they share any density reachability. A local subcluster is formed by a core LD and all of its retrieved nearby LDs within a predefined $d_{0}$ distance, and it expands using the two fundamental concepts of density-reachable and density-connected, as long as the conditions are met [51]:-$n_{j}$ is a member of a cluster $C_{i}$ if $n_{i}$ is a member of $C_{i}$ and $n_{j}$ is density-reachable from $n_{i}$.-$n_{i}$ and $n_{j}$ are density-connected if both of them are members of a cluster $C_{i}$.Assume the distance in the HSFZ between two sets $S_{1}$ and $S_{2}$ of LDs is defined as $dist\left(S_{1},S_{2}\right)=min\left\{dist\left(n_{i},n_{j}\right)|n_{i}\in S_{1},n_{j}\in S_{2}\right\}$. Two sets with at least the density of the thinnest cluster will be separated only if the distance between them is greater than $d_{0}$. The LoRa device density connection determines the maximum number of LDs $\left(LD_{max}\right)$ in an arbitrary cluster. Furthermore, in the steady-state phase, the GW is critical in computing the CHs of clusters by using network information from the IoT LoRa devices during network initialization.**Steady-state phase:** In this phase, the cluster head selection is performed using a naive Bayes classifier [48]. The GW plays a crucial role in generating the score value needed to categorize normal LoRa devices as CHs or cluster members. Based on the Bayes classifier, the GW determines the κ best LoRa devices to become CHs based on their binary score. The value of κ indicates the appropriate number of CHs. In [52], the authors explained what clustering algorithms were and how they worked. Moreover, there were three types of algorithms for choosing CHs: predetermined, random, and attribute-based. In the SFPCR scheme, we adopted attribute-based algorithms to select CHs. The GW constitutes the vector metric attributes $A^{\left(i\right)}$ for a LoRa device LD(i) to become the CH. $A^{\left(i\right)}$ = $\left\{A_{1}^{\left(i\right)},A_{2}^{\left(i\right)},\ldots A_{j}^{\left(i\right)}\right\}$, where *j* is the vector’s dimension. The attributes of an LD(i) include, but are not limited to, the residual energy ($E_{re\_N_{i}}\left(t\right)$), the link quality ($L^{\left(i\right)}$) between LD(i) and its neighbour, and the distance ratio $\left(D_{ratio}\left(n_{i},N_{i}\right)\right)$ evaluated as the minimum distance of a source node $n_{i}$ to the relay LoRa device $N_{i}$ in the LSFZ. As a result, a vector of attributes can be expressed as Equation (Equation 19).
(19)$$A^{\left(i\right)}=\left\{E_{re\_N_{i}}\left(t\right),L^{\left(i\right)},D_{ratio}\left(n_{i},N_{i}\right)\right\}$$Here, $E_{re\_N_{i}}\left(t\right)$ (Equation 24) and $L^{\left(i\right)}$ (Equation 13) are considered as the primary parameters, and $D_{ratio}\left(n_{i},N_{i}\right)$ is considered as the secondary parameter and is calculated using Equation (Equation 20).
(20)$$D_{ratio}\left(n_{i},N_{i}\right)={\left[\frac{d_{i}}{\sum_{i=1}^{N}\left[\frac{d_{i}}{N}\right]}\right]}^{min}$$In Equation (Equation 20), $n_{i}$ is the source LD in the *i*th cluster, $N_{i}$ is the selected relay LD at a time (t), and $d_{i}$ is the distance from the source LD to the relay LD, respectively. Then, the goal of the CH selection is to find an appropriate mapping relationship between *C* and $A^{\left(i\right)}$, where *C* is the binary score of attributes to which the LD belongs, either a cluster member (CM) or a cluster head (CH), and is expressed as $C=\{b_{0},b_{1}\}$.
(21)$$P\left(C=b_{1}|A^{\left(i\right)}\right)=\frac{P\left(A^{\left(i\right)}|C=b_{1}\right)P\left(C=b_{1}\right)}{P(A^{\left(i\right)})}$$$P(C=b_{1}|A^{\left(i\right)})$ expresses the probability of LD(i) becoming the CH with the classification $A^{\left(i\right)}$ using the Bayesian theorem expressed in Equation (Equation 21). Finally, the GW selects the LoRa device with the highest probability of becoming a CH within the clustered region based on Equation (Equation 22), which is also represented in Algorithm 2.
(22)$$C\leftarrow \underset{b_{1}}{\mathrm{argmax}}\;\left(P\left(A^{\left(i\right)}|C=b_{1}\right)\right)$$

In order to create connectivity between LoRa devices LD(i) in the HSFZ, each CH communicates with cluster members within a transmission radio range $\left(T_{r}\right)$ shown as $CH_{i}|d(LD\left(i\right),CH_{i}<T_{r}$. In addition, after a certain number of rounds at a time (t), the network carries out the clustering procedure again for reclustering. Once reclustering is complete, the GW continuously monitors the residual energy and link quality of LoRa devices in the HSFZ to choose new CHs based on the updated parameter values to ensure network continuity. The energy threshold $\left(E_{th}\right)$ is established to prevent the early death of some LoRa devices owing to their uneven energy decrease. In the SFPCR scheme, the energy threshold was assessed using the average energy of all LDs.
**Algorithm 2:** Pseudocode to select cluster heads in the HSFZ.**Input:**$X=X_{1},X_{2},\ldots ,X_{n}$: set of LoRa Devices (LDs)**Input:**$A^{\left(i\right)}$: LoRa Device metric attributes**Input:** N: number of LoRa Devices**Input:**κ: optimal number of CHs**Output:** Clusters of all LDs in HSFZ, CH allocation1:Network initialization2:All $C^{\left(i\right)}=b_{0}$3:**for**i←1: *N* **do**4:    Calculate the $E_{re\_N_{i}}\left(t\right),L^{\left(i\right)},D_{ratio}\left(n_{i},N_{i}\right)$5:    **if** $L^{\left(i\right)}\geq L_{th}$ && $E_{re\_N_{i}}\left(t\right)\geq E_{th}$ **then**6:        Add LD(i) to CH candidate set7:    **else**8:        Add LD(i) to cluster member (CM)9:    **end if**10:**end for**11:**for**AllLDs∈CHcandidateset**do**12:    Calculate the $P(C=b_{1}|A^{\left(i\right)})$13:    **if** $P\left(C=b_{1}|A^{\left(i\right)}\right)=\max\{P(C=b_{1}|A^{\left(i\right)}$)} **then**14:        $C^{\left(i\right)}=\left\{b_{1}\right\}$// LD(i) is set as a CH15:    **else**16:        $C^{\left(i\right)}=\left\{b_{0}\right\}$ // LD(i) is set as a CM17:    **end if**18:**end for**19:return CH

### 4.3. Relay LoRa Device Selection in the LSFZ

In this paper, we considered exploiting LDs in the LSFZ as relay devices to support reliable data transmissions from source LoRa devices in the HSFZ to the GW. To achieve a reliable relay device selection policy, the SFPCR scheme uses a centralized optimal relay assignment method that iteratively allows a single best relay LoRa device (LD) among the relay candidate set of LoRa devices $R=\{N_{1},N_{2},N_{3},\ldots ,N_{i}\}$ in the LSFZ, to access the channel and assist the bidirectional transmissions of all the distant LDs (represented as CMs) through the CH in the HSFZ.

The goal of our method was to devise a way to randomly select the best relay device to aid bidirectional communication between the source LDs (represented as CHs) and the GW, while taking into account two important performance parameters, namely, the link quality in terms of packet success probability and residual energy $E_{re\_N_{i}}\left(t\right)$ of the candidate relay LoRa devices. The $CH_{i}$ sends the relay a wake-on-radio (WOR) frame. This WOR frame wakes up the selected relay and sends packet information. The system model with $CH_{i}$ in the HSFZ acting on behalf of the source LDs is shown schematically in Figure 4. Meanwhile, the set *R* of potential candidate relays in the LSFZ includes LoRa devices, that is, $R=\{N_{1},N_{2},N_{3},\ldots ,N_{i}\}$. Our method compares the metrics to determine the best relay $N_{i}^{*}\epsilon R$. The packets collected by the $CH_{i}$ are transmitted to $N_{i}^{*}$ as the best relay from the candidate set because it meets the packet success probability as the primary evaluation parameter and residual energy threshold conditions for our relay device selection method. However, if any LD in the LSFZ does not meet the predefined threshold conditions, it cannot become a potential relay device.

We present four main phases to specify our centralized optimal relay assignment method. With the four main phases, our method iteratively selects an optimal relay LoRa device and uses it to achieve cooperative transmission by jointly considering a LoRa device’s link quality $\left(L_{N_{i}}\right)$ in terms of packet success probability and remaining energy information $E_{re\_N_{i}}\left(t\right)$. Initially, the residual energy of each LoRa device is critical in the relay selection process to avert excessive use of a single relay device on the appropriate channel. Thus, the concept of network fairness is not jeopardized in this way. Here, we explain the four phases in detail, respectively.

**Phase 1:** To create a set of potential relay devices, we employed the harmonic mean criterion to assess the link quality [24]. According to two-hop channel information, each LoRa device estimates the integrated link quality $L(N_{i})$ in terms of the packet success probability $(P_{suc}^{i}\left(loc\left(x\right)\right)$ based on the distance between them, the SNR, and the receiver sensitivity of the device. Then, $L(N_{i})$ can be shown in Equation (Equation 23).
(23)$$L\left(N_{i}\right)=2{\left(P_{suc}^{\left(S,N_{i}\right)}\left(loc\left(x\right)\right)+P_{suc}^{\left(N_{i},GW\right)}\left(loc\left(x\right)\right)\right)}^{-1}$$

For all LoRa devices that pass the first criteria, a candidate set of relay devices is constituted for further scrutiny in the succeeding phases in a bid to select the best relay LD.

**Phase 2:** The GW evaluates the residual energy $E_{re\_N_{i}}\left(t\right)$ of each candidate relay device $N_{i}$ at a given time (t) since its deployment in an IoT LoRa network. Assuming the total energy consumption $E_{total\_N_{i}}\left(t\right)$ used by a LoRa device $N_{i}$ during inactive and active modes at a particular time (t), $E_{0}$ is the initial energy of the LD. Then, the residual energy $E_{re\_N_{i}}\left(t\right)$ of the *i*th relay device is represented in Equation (Equation 24).
(24)$$E_{re\_N_{i}}\left(t\right)=E_{0}-E_{total\_N_{i}}\left(t\right)$$
where $E_{0}$ is the initial energy of the LoRa device at the initial time $\left(t_{0}\right)$, $E_{total\_N_{i}}\left(t\right)$ is the energy dissipated by the LoRa device during inactive mode at time (*t*) and active mode.

**Phase 3:** We used a positive weight factor (0≤α≤1) that indicated the preference between the integrated link quality α and the residual energy weight (1−α) for all $N_{i}\epsilon N$ LoRa devices. Equation (Equation 25) indicates the combined relay parameters $W(N_{i})$.
(25)$$W\left(N_{i}\right)=\alpha \left(E_{re\_N_{i}}\left(t\right)\right)+\left(1-\alpha \right)L\left(N_{i}\right)$$

$L(N_{i})$ is the integrated link quality in terms of the LoRa device’s packet success probability, $E_{th}$ is the threshold energy considered to be the average residual energy and $L_{th}$ represents the threshold link quality. Given, the following conditions, $E_{th}\leq E_{re\_N_{i}}\left(t\right)\leq 1$ and $L_{th}\leq L\left(N_{i}\right)\leq 1$ hold true.

**Phase 4:** By the ranking method, when a candidate LoRa device $N_{i}$ has the highest rank as $R_{N_{i}}^{*}$ among all the candidate LDs by evaluating with the formula depicted as the Equation (Equation 26), it becomes the best relay device.
(26)$$R_{N_{i}}^{*}=\underset{i\in N}{arg max}W\left(N_{i}\right)$$
where $R_{N_{i}}^{*}$ means the candidate $N_{i}$ LoRa device with the highest rank that becomes the best relay device to transmit data from the source LD to the GW. Since the relay selection process is iterative, the rest of the devices back off and wait for the next transmission round.

## 5. Performance Evaluation

In this section, we evaluate the performance of the proposed SFPCR scheme through simulations. We compare the SFPCR scheme with three existing schemes: the FSRC scheme [33], the cluster-based scheme [21], and the mesh scheme [37] for performance evaluation. To do this, we first describe our simulation environments and scenarios and next present their performance evaluation metrics. Last, we explain the performance comparison of the FSRC and the three existing schemes by simulation results.

### 5.1. Simulation Environments and Scenario

To validate the SFPCR scheme’s performance, an NS-3-based simulation environment was used to model multihop communications in the LoRa network. The performance of the FSRC scheme, the cluster-based scheme, and the mesh scheme were compared to the SFPCR scheme based on the relay installation and selection procedure. The NS-3 simulation tool is suitable for large networks anchored on discrete event simulation models of a system that changes to its state occurrence at discrete points in the simulation time [53].

Except for the GW, all the devices had the same communication needs. Each LoRa device sent out data packets on a regular basis, and transmissions only happened during certain time slots to avoid collisions. LoRa devices always kept their duty cycle below 1%, which is the minimum amount allowed by regulations [4]. Different spreading factors were spread out evenly based on how far away the devices were from the GW.

Table 2 lists the simulation parameters used in our model. We considered a circular network region with a cell radius of R=6 km and a GW in its centre to handle communication. The LoRa devices’ distance from the GW was based on a Gaussian distribution. The parameters bandwidth, spreading factor, transmission power, and code rate were assigned to each device. In addition, each device transmitted an average of 20 bytes every packet. The estimated transmit power could be set from 2 dBm to 14 dBm, the channel bandwidth was 125≤BW≤250 kHz, and the channel code rate was 4/5. It is hard to say how many data the GW can handle, but it is safe to presume that more data provided more often will limit how many devices it can interface with. We compared the performance of two existing schemes to see which one was better. We also investigated the differences between them to see if our SFPCR scheme would be better based on the performance metrics.

### 5.2. Performance Evaluation Metrics

The metrics used for the performance evaluation of the proposed SFPCR scheme and the existing schemes [21,33,37] were: packet success probability, energy consumption, and the number of packets. In simple terms:Packet success probabilityA frame transmission is considered successful when no collisions occur and all the bits in the frame are accurately decoded despite interference [54]. The harsh environment and capture effects are two factors that can have an impact on the success of data transmissions between LDs and GWs. It is critical to consider both the likelihood of the GW receiving an uplink from the source LD and the likelihood of the device successfully receiving a downlink.Energy consumptionWe considered the energy consumption among LoRa devices, which refers to the number of delivered data bits per unit of energy consumed by a LoRa device [39]. The SFPCR scheme aimed to balance the HSFZ’s energy efficiency by shortening the ToA of the LDs.Number of PacketsWhen analysing traffic behaviour in a LoRa network, the number of packets passing through the network to the GW is critical. In the LoRaWAN network, packets are sent infrequently and are affected by a variety of factors. As a result, the primary objective here was to ensure an effective transmission that limited duplicate packets, collisions, and retransmissions, all of which produce large quantities of network congestion and thus impede overall network performance.

### 5.3. Simulation Results

In this section, we present the IoT LoRa performance evaluation using the metrics from Table 2. We compared the proposed SFPCR scheme to the FSRC scheme, also called the relay scheme, the mesh scheme, and the cluster-Based scheme. We focused on three factors: transmission probability, packets sent by a given number of LoRa nodes, and energy consumption. The number of nodes, payload size, and gateway distance were compared. This study validated the suggested technique for multihop LoRa networks by using relay and clustering approaches in corresponding spreading factor zones.

Figure 5, Figure 6 and Figure 7 illustrate how the packet success probability of LoRa devices varied with the number of nodes, payload size (B), and node distance from the GW (m). All the schemes exhibited an inverse proportionality between the success probability and the variables of number of nodes, payload size, and node distance from the GW. Figure 5 indicates that increasing the number of nodes decreased the chance of packets successfully reaching the GW for the FSRC, cluster-based, and mesh schemes. This was due to the capture effect as nodes competed for the channel and collisions arising from blind transmissions as these nodes transmitted data packets as soon as they were ready to transmit [55], which hindered the successful delivery of packets to the GW. In contrast, the proposed SFPCR scheme employed collision-free transmission [45]. Due to this regulated access to the medium, each LoRa device transmitted its packets directly to the selected recipient. The SFPCR scheme deployed, among other things, density-based subspace clusters in the HSFZ and relay nodes in the LSFZ in order to boost and improve the performance of remote nodes so that data packets could be successfully sent to the GW with no transmission errors. Consequently, the SFPCR scheme outperformed the other schemes with the highest number of LDs by approximately 47.6% packet success probability, followed by the FSRC scheme at 24.7%, as shown in Table 3. However, as the number of LDs increased, the performance degraded because an increase in the number of nodes caused an automatic increase in the probability of packet error and interference [56].

Figure 6 depicts an inverse proportionality between success probability and payload size (B) for the SFPCR scheme and other comparative schemes. Because the LoRa network is utilized for long-distance transmission, an increase in packet size has a detrimental impact on the network’s performance. Therefore, the smaller the size of a data packet, the greater the possibility of success. The success probability of the SFPCR scheme was roughly 65.7% with a default payload size of 20 bytes, as shown in Table 3. It provided a superior method for reducing the quantity of traffic delivered to the GW by using data aggregation for all nodes clustered in the HSFZ, in contrast to previous techniques.

Figure 7 demonstrates a correlation between the distance of LoRa devices from the GW and their success probability. The success probability of all schemes diminished exponentially with rising average distances. However, at a distance between 3000 m and 3500 m, the SFPCR scheme and the FSRC scheme improved the success probability marginally due to the use of relays to extend packets from far LoRa devices to the GW. Moreover, the SFPCR scheme outperformed the other schemes in terms of success probability by more than 29.1% as the transmission distance from the GW increased. The usage of subspace clusters in the HSFZ and relays with success probabilities above the threshold in the LSFZ to transfer packets from far LDs to the GW mitigated the near–far fairness issue. The other schemes, on the other hand, demonstrated a negative decrease in success probability as the distance between the LDs and the GW increased due to poor link quality, noise, collision among the nodes, and other interference from obstacles such as buildings and vegetation encountered by nodes located far from the GW [55].

Figure 8 illustrates the variation in the number of nodes as a function of the energy consumption for the SFPCR scheme and the three other schemes under consideration. In contrast to the SFPCR scheme, the results for all three comparative schemes demonstrated a slightly direct correlation between energy use and number of nodes. The FSRC scheme utilized the most energy, exceeding 29 μJ, as a result of using more LDs with a higher SF and broadcasting packets sent throughout the network, which introduced considerable routing overhead and used a great deal of energy. Similar to the cluster-based scheme, which had huge cluster sizes, the cluster member nodes had to connect with other nodes located at great distances, resulting in increased energy consumption owing to intracluster distance communication. In addition, the mesh scheme consumed 22.2% more energy than the SFPCR scheme because of the unnecessary multihops to send a signal strong enough to reach the GW. However, the SFPCR scheme consumed approximately 67.4% less energy than the comparative schemes because it primarily utilized short hops with a lower SF, controlled redundant transmission by data fusion from CHs in the HSFZ and also managed the number of transmissions and retransmissions to the GW by employing slotted ALOHA [45]. Consequently, the network’s lifespan was increased.

Figure 9 shows a direct proportionality between payload sizes and energy consumption for all schemes. As packet size increases, so does energy consumption, limiting the lifespan of an IoT network that is energy constrained. This is due to the broadcast storm problem caused by source nodes and relay nodes deployed in the network; in comparison, the cluster-based, mesh, and proposed SFPCR schemes consumed relatively little energy, limiting the use of distant nodes with higher SF to transmit directly to the GW, which consumes a great deal of energy. In addition, the SFPCR scheme transmitted small packets with short hops and a lower spreading factor in order to save transmission power. This implied that the source node in the HSFZ transmitted to the closest CH and relay node, thereby reducing the amount of time spent in the air. Consequently, the amount of energy lost was restricted.

In Figure 10, all schemes demonstrated a low quantity of energy consumption between the GW and approximately 3000 m, where direct transmission occurred; nevertheless, the FSRC scheme lost significantly more energy than the other schemes because of broadcasts made by relays and source nodes. The results demonstrated that the proposed SFPCR scheme consumed the least amount of energy as the distance increased, and the number of transmissions was regulated because all distant nodes in the HSFZ coordinated their transmission through the CHs rather than a direct transmission of data packets to the GW and because of the use of relay nodes in the LSFZ with a low SF to transmit packets of nodes with a success probability below the threshold.

As the number of nodes in the network increased, Figure 11 shows that the variation in the number of packets remained essentially constant across all the different schemes. Following the cluster-based and mesh schemes in terms of the number of packets used for varying numbers of nodes in the IoT sensor area comes the newly proposed SFPCR scheme, which used the fewest packets overall. The FSRC Scheme, on the other hand, had the most packets since it broadcast packets from the source nodes, relay nodes, and normal nodes. This caused a redundancy in the network as a result of the increased number of duplicate packets that were broadcast.

Similarly, Figure 12 demonstrates that the SFPCR scheme forwarded the fewest data packets to the GW. Furthermore, in the SFPCR scheme, the number of sent packets was more important than their size. Consequently, there was always a limit at which increasing the packet size would no longer increase performance and may instead decrease it. Cluster-based and mesh schemes had roughly the same number of packets as the payload size grew because their topologies were nearly identical. In contrast to the SFPCR scheme, the broadcast phase of the FSRC scheme resulted in a large number of duplicate packets in the network.

As depicted in Figure 13, the SFPCR scheme demonstrated a single packet transmission at near ranges from the GW to approximately 3000 m, beyond which the number of packets increased modestly due to the employment of relay nodes in the LSFZ. In contrast, the CHs in the HSFZ aggregated data packets from all remote source nodes within the HSFZ, hence lowering network traffic flow. Similarly, the FSRC scheme from the GW to a distance of approximately 3000 m demonstrated a single packet transfer because no relay nodes were utilized in that zone. Beyond this stage, relay nodes that duplicated the sent packets were deployed. In the mesh and cluster-based schemes, each node obtained data from one node while passing data to the next node, resulting in an increase in packets. Consequently, as the distance from the GW rose, so did the number of node connections and hops.

## 6. Conclusions and Discussion

In this paper, a novel SF-partition-based clustering and relaying scheme was proposed to solve the near–far fairness problem in IoT LoRa networks. The proposed SFPCR scheme utilized clustering and a multi-hop relay approach with the aid of low-SF LDs to increase the minimum success probability of distant nodes with a high SF. In the proposed scheme, the SF-based network partitioning method was used to determine the best partitioning threshold point for dividing SF zones into an LSFZ and an HSFZ to bridge the performance gap among SF zones by using an optimal SF division based on a heuristic algorithm. Then, by a combination of the DBSCAN algorithm and naive Bayes classifier, the density-based subspace clustering method constructed clusters of arbitrary shape for adjacent LDs in the HSFZ and selected CHs by a binary score representation. To extend the transmissions of CHs in the HSFZ by multihop communications, the relay LD selecting method chose the best relay LD in the LSFZ by using the harmonic mean based on the packet success probability and the remaining energy. The packet success probability of each LD in a certain SF zone was examined based on the hop distance of each hop to provide the optimal zone allocation with connection fairness. Through comprehensive simulations, we revealed that the SFPCR scheme exhibited the highest packet success probability of 29.1% for distant LDs at 6000 m, followed by the FSRC scheme at 1.12%, the mesh scheme at 0.67%, and lastly the cluster-based scheme at a negligible value of 0.036%. Using the default payload size of 20 bytes, the proposed SFPCR scheme also exhibited the highest success probability of 65.7%, followed by the FSRC scheme at 44.6%, the mesh scheme at 34.2%, and lastly the cluster-based scheme at 29.4%. The SFPCR conserved more energy for the LDs in order to prolong the lifetime of IoT LoRa networks in comparison to previous LoRa multihop schemes. Future work will include the development of intelligent clustering algorithms that can maintain up-to-date information on the link quality between LDs and their neighbours and react to sudden topological changes, as well as the responsiveness to changing network conditions through the use of lightweight reinforcement learning techniques.

## Figures and Tables

**Figure 1 sensors-22-09332-f001:**
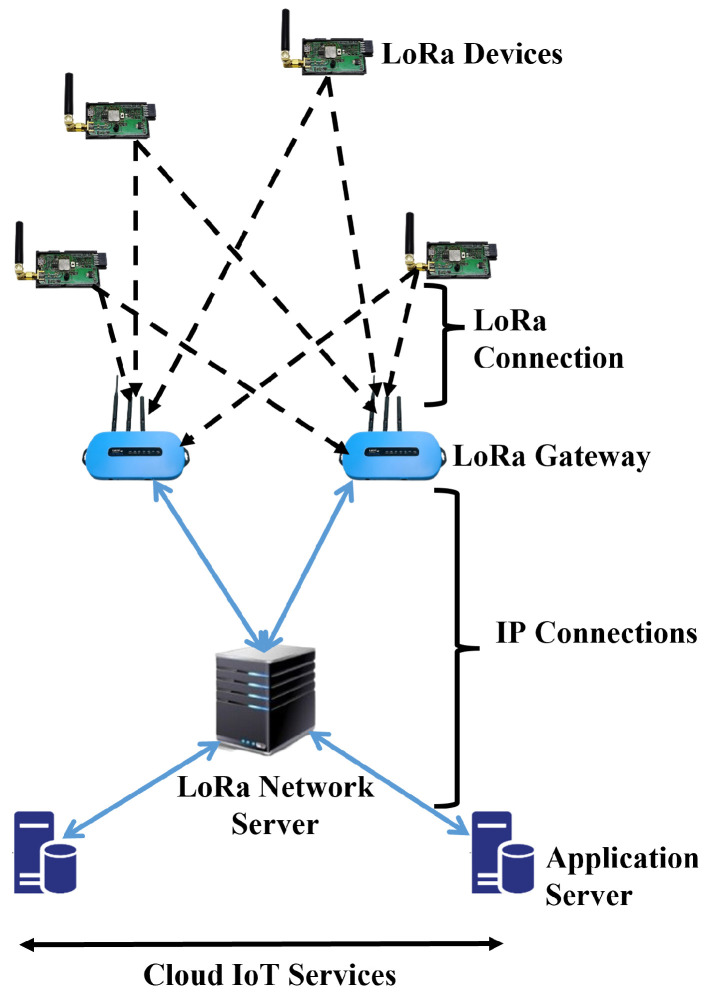
General network architecture for IoT LoRa networks.

**Figure 2 sensors-22-09332-f002:**
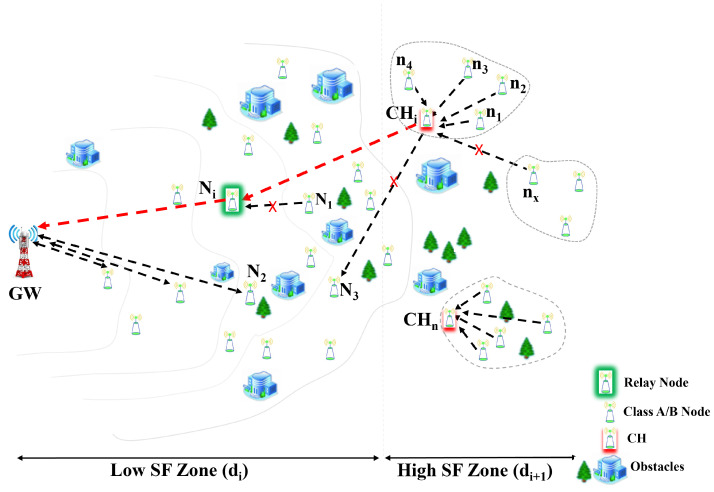
Model of the relay and cluster process in the proposed SFPCR scheme.

**Figure 3 sensors-22-09332-f003:**
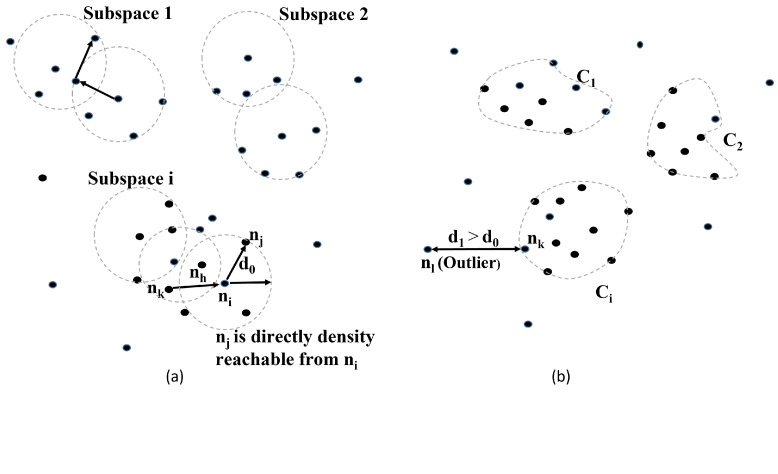
Arbitrary subspace clusters in the HSFZ: (**a**) Node density-reachability, (**b**) Node density-connected arbitrary subspace clusters.

**Figure 4 sensors-22-09332-f004:**
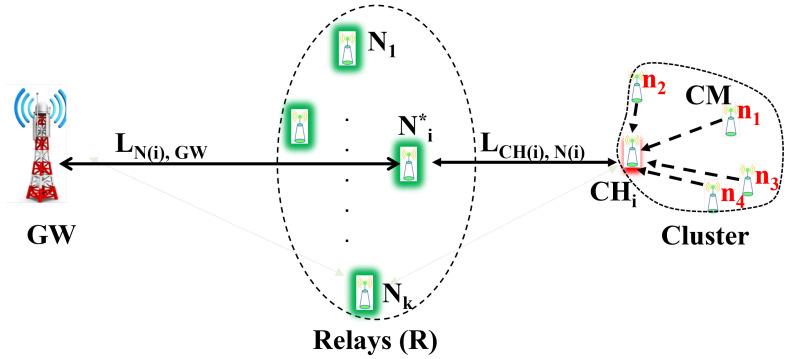
The relay selection model.

**Figure 5 sensors-22-09332-f005:**
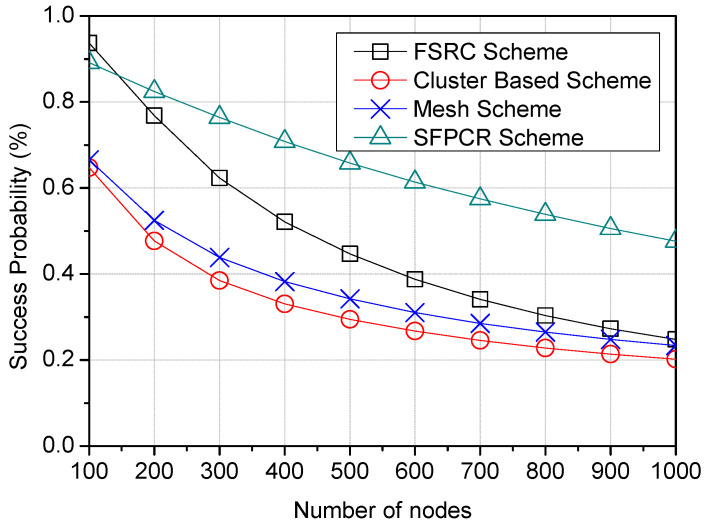
Plots variation in Success Probability according to the number of nodes in the network with a comparison between the SFPCR scheme and other protocols.

**Figure 6 sensors-22-09332-f006:**
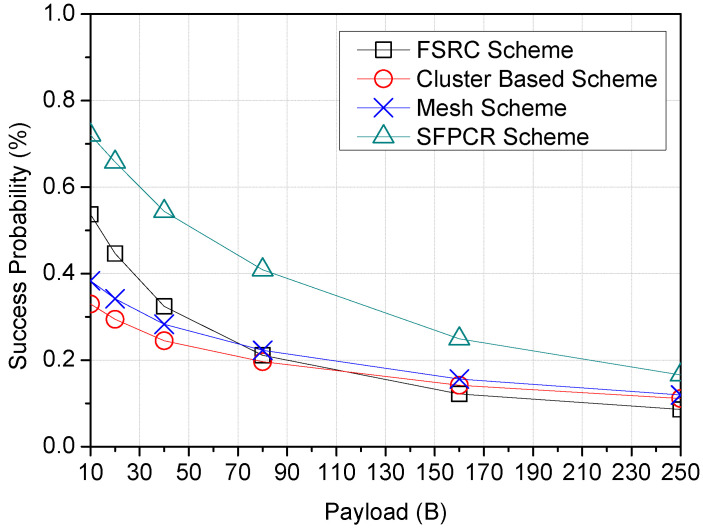
Plots variation in Success Probability according to the Payload Size (B) with a comparison between the SFPCR scheme and other protocols.

**Figure 7 sensors-22-09332-f007:**
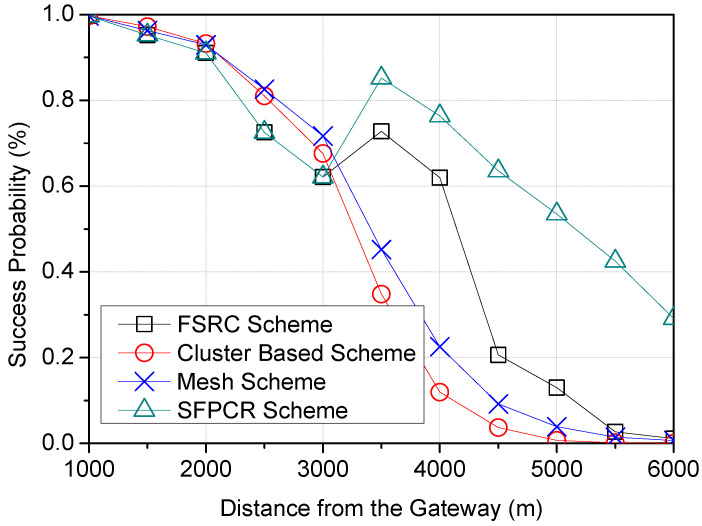
Plots variation in Success Probability according to the Distance of LoRa devices from the GW with a comparison between the SFPCR scheme and other protocols.

**Figure 8 sensors-22-09332-f008:**
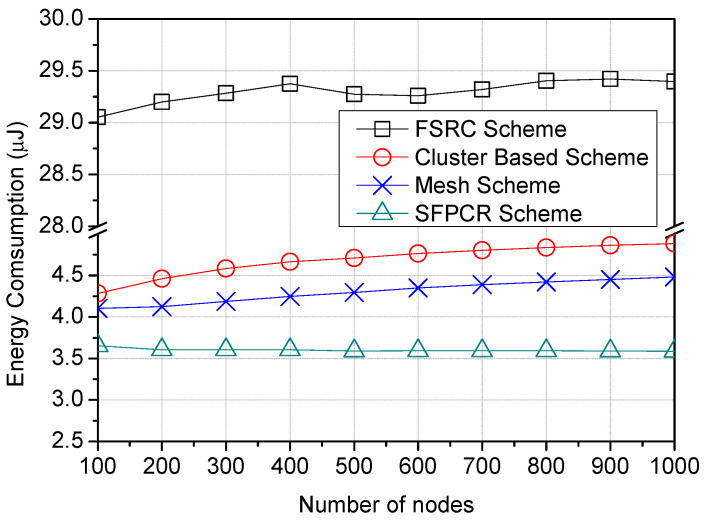
Plots variation of energy consumption (μJ) as a function of number of nodes with a comparison between the proposed SFPCR scheme and other protocols.

**Figure 9 sensors-22-09332-f009:**
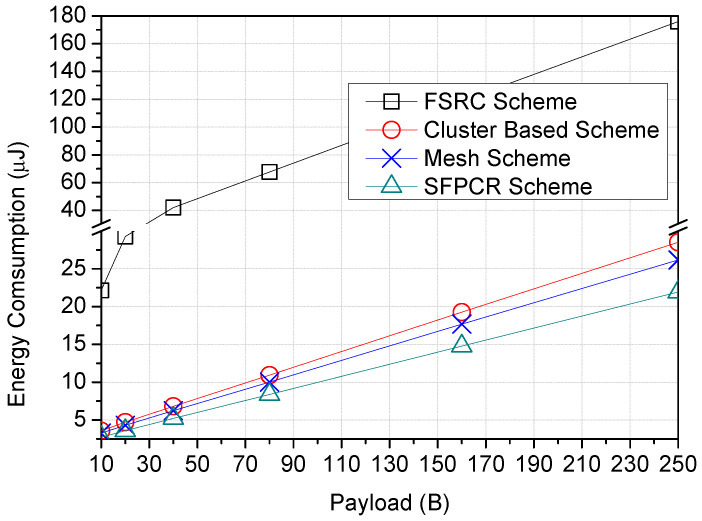
Energy consumption as a function of payload size (B) with a comparison between the proposed SFPCR scheme and other protocols.

**Figure 10 sensors-22-09332-f010:**
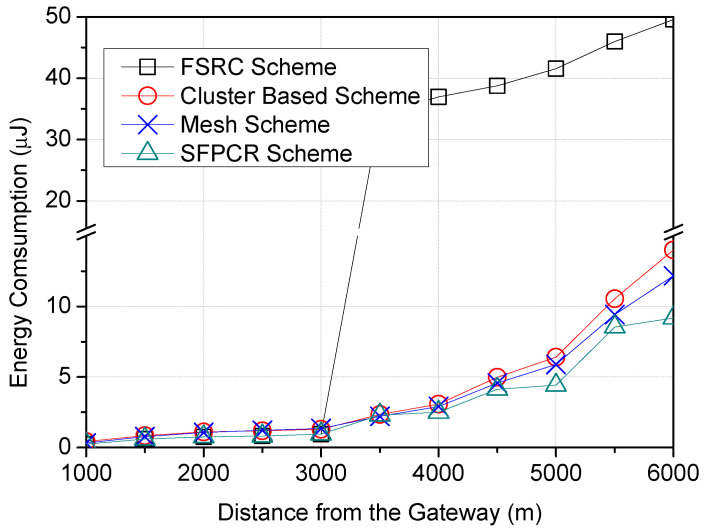
Energy consumption as a function of LoRa devices’ distance from the GW (m) with a comparison between the proposed SFPCR scheme and other protocols.

**Figure 11 sensors-22-09332-f011:**
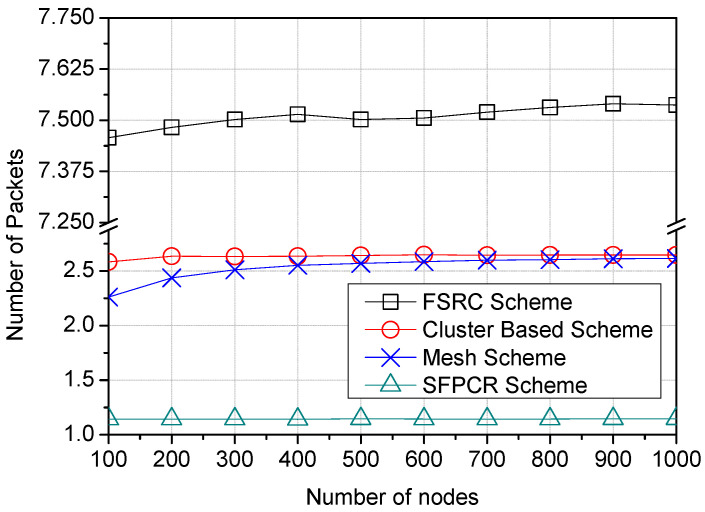
Variation in the number of packets according to the number of nodes in the network with a comparison between the proposed SFPCR scheme and other protocols.

**Figure 12 sensors-22-09332-f012:**
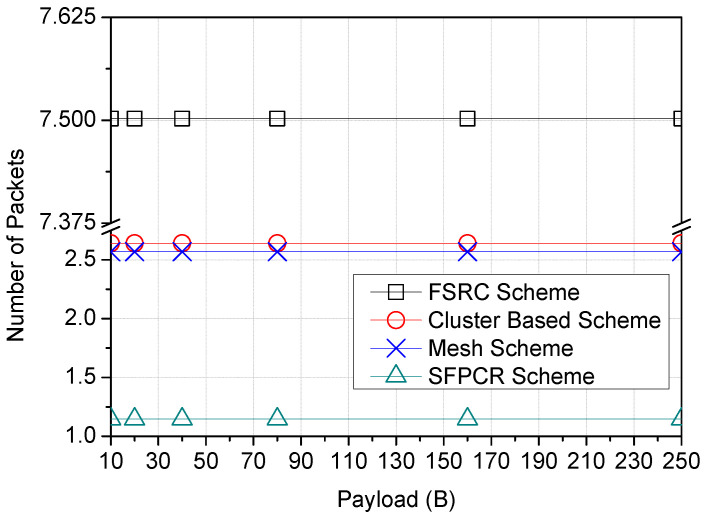
Variation in the number of packets according to the payload size (B) with a comparison between the proposed SFPCR scheme and other protocols.

**Figure 13 sensors-22-09332-f013:**
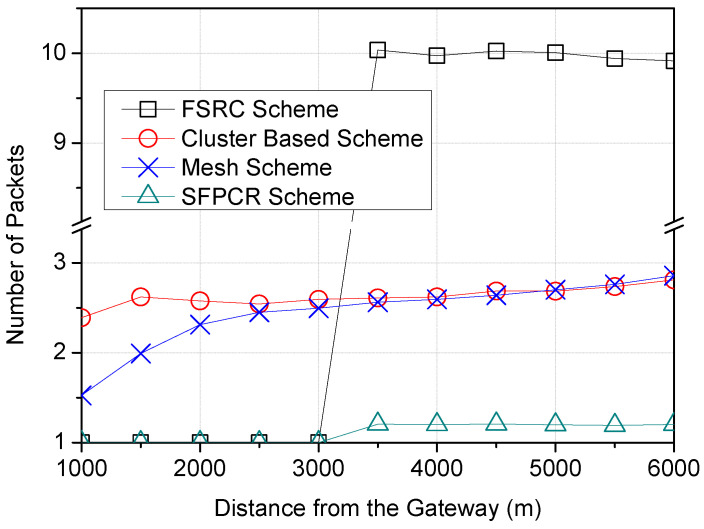
Variation in the number of packets according to the the distance of LoRa devices from the GW with a comparison between the proposed SFPCR scheme and other protocols.

**Table 1 sensors-22-09332-t001:** LoRa parameters based on SF.

SF	SNR	Receiver Sensitivity	Bit Rates
(i)	(dB)	qs (dBm)	(kbps)
7	−6	−123	5.47
8	−9	−126	3.13
9	−12	−129	1.76
10	−15	−132	0.98
11	−17.5	−135.5	0.54
12	−20	−137	0.29

**Table 2 sensors-22-09332-t002:** Simulation parameters.

Parameter	Values
Cell radius (R)	6 km
Number of LDs (N)	100–1000
Spreading factor (SF)	7–12
Bandwidth	125,250 (kHz)
Payload length	10–250 (Bytes)
Transmission power	14 dBm
Data rate	0.25–5.47 (kbps)
Coding Rate	4/5
Simulation time	3600 s
Payload CRC	ON
Duty cycle regulation	1 %
Channel frequency	868 (MHz)
Header (H)	0
Preamble symbol	8
Packet interval rate	10–15 (min)
Low data rate optimization (DE)	1
Path loss exponent (γ)	4

**Table 3 sensors-22-09332-t003:** Simulation results.

Success Probability (%)				
**Scheme**	**FSRC**	**Cluster Based**	**Mesh**	**Proposed SFPCR**
Number of nodes (1000)	24.7	20.1	23.3	47.6
Default payload size (20 bytes)	44.6	29.4	34.2	65.7
Distance from the GW (6000 m)	1.12	0.036	0.67	29.1

## Data Availability

Not applicable.

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
