# Peer review of "SF-Partition-Based Clustering and Relaying Scheme for Resolving Near–Far Unfairness in IoT Multihop LoRa Networks"

_sensors, 2022, doi:10.3390/s22239332_

Round 1

Reviewer 1 Report

In this paper, authors have proposed a novel clustering scheme based on SF partitioning to solve near-farness problem in LoRa networks. Overall, the paper is well written.

1.     References [6], [10], [19], [41] are not cited anywhere in the manuscript.

2.     Few references are so old and incomplete, correct them and replace them suitably.

3.     Proofread the paper. Some sentences are required rephrasing and restructuring for better understanding.

4.     Correct the symbol of “belongs to” in Line 3 of Section 3.1 and in other places as well.

5.     What is the meaning of intensity (lambda) of LoRa devices?

6.     What is the rationale behind choosing the value of k = 4 in initial set up phase of section 4.2?

7.     What is the complexity of your proposed cluster head selection mechanism?

8.     Why you have chosen FSRC scheme [35], the Cluster-based scheme [22], and the Mesh scheme [39] for comparison?

9.     It is advisable to include a table in Section 5 showing percentage improvement of your proposed work as compared to existing works.

1-  What is/are the limitation(s) of your proposed work?

1- How proposed mechanism prevents the selection of low energy node as cluster head? How will it take care of energy hole problem in the network?

Reviewer 2 Report

This manuscript tackles the near-far fairness problem in IoT Multi-Hop LoRa Networks and then propose a novel scheme to achieve enormous LD connectivity with fairness in IoT multi-hop LoRa networks. The manuscript is clear and presented in a well-structured manner.

On the other hand, the paper should be revised by considering the following issues:

MAJOR ISSUES

+Most of the cited references are mostly recent publications (within the last 5 years) and relevant. On the other hand, there are some references dated two decades ago. Some of them can be updated.

+ The number of references are good. On the other hand, the bibliography can still be improved. Different approaches on similar problems should be considered in this paper. The following two papers considered a similar problem in multi-hop IoT network which consists of sensors, robots and a UAV with limited battery capacity. For this purpose, I strongly recommend the authors should include the following two papers in their related work in order to clarify not only the main contribution but also motivation of proposed approach in this paper in the related literature.

- O. M. Gul, A. M. Erkmen, "Energy-Efficient Cluster-Based Data Collection by a UAV with a Limited-Capacity Battery in Robotic Wireless Sensor Networks", Sensors, vol.20, no.20,Sep. 2020.

+The proposed clustering scheme performs well. The motivation behind it should be explained better.

+ How can Equation (3) be derived frrom Equation (2)?

+In Equation (6), Is "E_s" term same for all i? If "E_s" term is same for all i, the last Term can be converted into "n*E_s". Otherwise, "E_s" should include some index "i".

+The left side of Equation (13), P_avg function should also include "x" or "loc(x)" as parameter.

+The contributions and organization of the paper should be clearly given as a separate subsections in  the introduction section.

+Tables are very clear and informative. On the other hand, Table 1 should be explained better by adding more information to its caption.

+Most of the figures/schemes are clear. They show the data properly. It is not difficult to interpret and understand them. On the other hand, Figure 3 should be explained better by adding more information to its caption.

+ The conclusion should give the key results and main contributions more clearly. By the way,  "Through comprehensive simulations, we proved that ...in this section does not make sense. Simulations do not prove anything, they can show something.

+ Future work should part in the conclusion section should be improved.

MINOR ISSUES

+The grammatical errors and typos should be fixed.

+Equation (3) and (12) should be given in single line.

+The references in the bibliography should be given in the same style. The following link should be checked: https://www.mdpi.com/authors/references

Reviewer 3 Report

This is a good paper with an interesting approach and some original contributions. The paper is well structured and has enough experimental evaluation results. 

However, the presentation of the results needs to be improved and better explained. The simulation scenario is not clearly explained. For example it says network size 12 km. What is the size? Lineal network, area? 12 km^2. The number of end-devices varies from 100 to 1000 but the authors indicate in the introduction that the clustering method proposed was for large networks. 1000 end-devices within a 12 km network is not a large network. There are other minor problems like indicating in the figures for example Node density when it is Number of nodes in the network. A density is basically the relation between a quantity an a unit of area o volume. 

There is no explanation on how LoRa nodes would implement the clustering proposed. What kind of messages nodes exchange before being connected, who starts the process, if it is dynamic. I guess there are several loose ends there that should be carefully revised before accepting the paper for publication.

Round 2

Reviewer 2 Report

The paper is acceptable in its current form. Minor spell check is suggested.

Reviewer 3 Report

The paper is now ready to be published